# Evaluating diagnostic accuracies of Panbio™ test and RT-PCR for the detection of SARS-CoV-2 in Addis Ababa, Ethiopia using Bayesian Latent-Class Models (BLCM)

**Abay Sisay** [1,2]☯*, **Sonja Hartnack** [3]☯, **Abebaw Tiruneh** [1,4], **Yasin Desalegn** [4], **Abraham Tesfaye** [1,5], **Adey Feleke Desta** [2]

1 Department of Medical Laboratory Sciences, College of Health Sciences, Addis Ababa University, Addis Ababa, Ethiopia, 2 Department of Microbial, Cellular and Molecular Biology, College of Natural and Computational Sciences, Addis Ababa University, Addis Ababa, Ethiopia, 3 Section of Epidemiology, Vetsuisse Faculty, University of Zurich, Zurich, Switzerland, 4 Addis Ababa Public Health Research and Emergency Management Laboratory, Addis Ababa Health Bureau, Addis Ababa, Ethiopia, 5 Diagnostic Unit, Center for Innovative Drug Development and Therapeutic Trials for Africa, CDT-Africa, Addis Ababa, Ethiopia

☯ These authors contributed equally to this work.
* abusis27@gmail.com, abay.sisay@aau.edu.et

**Data Availability Statement:** All data set used in this study are included in this manuscript.

## Abstract

### Background

Rapid diagnostics are vital for curving the transmission and control of the COVID-19 pandemic. Although many commercially available antigen-based rapid diagnostic tests (Ag-RDTs) for the detection of SARS-CoV-2 are recommended by the WHO, their diagnostic performance has not yet been assessed in Ethiopia. So far, the vast majority of studies assessing diagnostic accuracies of rapid antigen tests considered RT-PCR as a reference standard, which inevitably leads to bias when RT-PCR is not 100% sensitive and specific. Thus, this study aimed to evaluate the diagnostic performance of Panbio™ jointly with the RT-PCR for the detection of SARS-CoV-2.

### Methods

A prospective cross-sectional study was done from July to September 2021 in Addis Ababa, Ethiopia, during the third wave of the pandemic involving two health centers and two hospitals. Diagnostic sensitivity and specificity of Panbio™ and RT-PCR were obtained using Bayesian Latent-Class Models (BLCM).

### Results

438 COVID-19 presumptive clients were enrolled, 239 (54.6%) were females, of whom 196 (44.7%) had a positive RT-PCR and 158 (36.1%) were Panbio™ positive. The Panbio™ and RT-PCR had a sensitivity (95% CrI) of 99.6 (98.4–100) %, 89.3 (83.2–97.6) % and specificity (95% CrI) of 93.4 (82.3–100) %, and 99.1 (97.5–100) %, respectively. Most of the

**Funding:** This work was financed by Addis Ababa University through adaptive research and problem solving project, with project title, Evaluation and Validation of the Diagnostic Performance of SARS-CoV-2 rapid test for the detection of Novel Corona Virus, Ref #-PR/5.15/590/12/20 and no external funding was received. The funders anywhere not involved in study design, data collection and analysis, decision to publish, or preparation of the manuscript. The contents are purely the responsibilities of the authors and did not represent and reflect the view of the funder.

**Competing interests:** We, the authors declare that they have no known competing financial interests or personal relationships that could have appeared to influence the work reported in this paper

study participants, 318 (72.6%) exhibited COVID-19 symptoms; the most reported was cough 191 (43.6%).

## Conclusion

As expected the RT-PCR performed very well with a near-perfect specificity and a high, but not perfect sensitivity. The diagnostic performance of Panbio™ is coherent with the WHO established criteria of having a sensitivity ≥80% for Ag-RDTs. Both tests displayed high diagnostic accuracies in patients with and without symptoms. Hence, we recommend the use of the Panbio™ for both symptomatic and asymptomatic individuals in clinical settings for screening purposes.

## Introduction

COVID-19 was first detected in December 2019, declared a public health emergency in January 2020, and then categorized as a pandemic in March 2020. Since then, numerous new diagnostic tests for COVID-19 have been developed in a very short time to reliably identify infected patients and tackle its future spread of the disease [1–3].

In Africa, the current state of the health system and laboratory diagnostic capacities are limited concerning managing outbreaks as early as possible and reducing the burden of disease successfully. This hampers realizing the 2030 SDG with lots of pitfalls in the diagnostic capacity and with so many people failing to get diagnosed [4, 5].

Ethiopia, a low-income country, faces a lack of trained laboratory personnel and material resources. Amongst the African countries, Ethiopia ranked 6th with 468 985 COVID-19 cases as of 15 Feb 2022. The third and fourth waves are characterized by rapid transmission and a high positivity rate [6, 7]. To tackle the pandemic, a central command public health emergency operation center (PHEOC) was implemented and the challenges and achievements in Ethiopia have been described [8, 9].

The current choice of established tests for the diagnosis of COVID-19 with a high diagnostic performance using respiratory swab samples is RT-PCR. Unfortunately, the costs and the infrastructure demand for most laboratories in low-income countries, like Ethiopia, are too much, thus excluding a wide-spread usage of RT-PCR. Alternative reliable testing modalities, being convenient approaches to reach more clients and making healthcare services accessible, are needed. Rapid diagnostic tests may potentially be components of a successful COVID-19 disease control strategy by promptly identifying cases at lower costs, which would ultimately lead to saving more human lives [4]. A number of RDTs have been developed and commercialized [4, 8, 10], and the WHO has established criteria and recommends as "the use of Ag-RDTs that meet minimum performance requirements of ≥ 80% sensitivity and ≥ 97% specificity" prior to use [11].

Rapid diagnostic testing (RDT) has become a game-changer for triaging patients and crucial medical decisions [8, 12]. The impact of RDT has been significant because results can be delivered in a short turnaround time of COVID-19 testing. Despite the ease of application and low cost, RDTs are still in need of attention on their quality diagnostic performance for the containment of the virus. On top of these, compared to RDT, the procedure of RT-PCR is sophisticated and may lead to specimen contamination [13].

At the national level, since May 2021, Ethiopia has started using the Panbio™ test for the diagnosis of COVID-19 in line with RT-PCR after checking its suitability in the selected health

facility as a pilot study. As of 8 August 2022, a total of 5,158,430 COVID-19 tests were performed, of which 1,128,391 were done by rapid antigen test kits [14–16]. Yet, there has been no documented evidence of its diagnostic field performance in Ethiopia.

Moreover, a lot of studies are available elsewhere that evaluated Panbio™ against RT-PCR as the reference standard [17–20], which inevitably leads to bias. Yet, to our knowledge, no study has been conducted using Bayesian latent class models (BLCM) to assess the diagnostic performance of Panbio™. Although latent class models to estimate diagnostic test accuracies in the absence of a perfect reference standard, also called gold standard, have been proposed decades ago by Hui and Walter [21], the world organization for animal health (OIE) endorsed Bayesian latent class models in the context of diagnostic test evaluation in 2013 [22] and reporting STARD guidelines specific for BLCMs have been developed [23], there is a scarcity in human medical applications. Potential reasons for this may be their complexity requiring expert statistical knowledge in the analysis and the interpretation of the results [24].

Additionally, with regard to the translation of BLCM results into clinical practice, Bayesian latent class models do not rely on a clinical definition of a target condition, but on a statistical one. Here, the assumption is that a positive test result of the first test indicates the same condition as a positive test result of a second test. In reality, the detection of ribonucleic acids by RT-PCR, which might be leftovers from a past infection, does not necessarily equate the meaning of a positive test result in a rapid antigen test detecting the presence of viral capsid proteins. Thus, the latent state assumed by the BLCM is the presence of viral RNA and antigens in the samples rather than "individual is infected with the virus" [25]. However, this limitation does not only pertain to BLCMs but also to the classical approach of determining diagnostic test accuracies of a rapid antigen test for example, by comparing it with RT-PCR considered as a perfect reference standard.

For COVID-19, the shortcomings of RT-PCR have been described—the potential occurrence of false positive and false negative test results [26–29], which potentially invalidate diagnostic accuracy studies. We have tested 279 patient samples with three RT-PCRs in Ethiopia and the test results were not perfectly congruent, indicating that not all RT-PCRs are 100% sensitive and specific [30]. The so-called 'reference standard error bias' indicates the problem that if the reference test (here the RT-PCR) results are wrongly classified as positive or negative, the sensitivities or specificities of the new test under evaluation will be under or overestimated, since all misclassifications will only be attributed to the new test.

In BLCM models, none of the tests is considered a perfect reference standard [22]. In contrast, the sensitivity and specificity of all tests are estimated based on the frequencies of the cross-classified test results. Latent means that the true status of each individual is not observed directly, but can be obtained from the information contained in the data. When evaluating diagnostic tests with a BLCM approach, a "test" comprises the entire process from taking the sample, transporting, and any pre-processing steps as well as applying the test in question. Throughout this process, other sources of variation may occur, which are not present if solely the (analytical) sensitivity and specificity are considered under laboratory conditions, which are well controlled or even ideal but are not representative of real-world testing [31].

Thus, this study aimed to assess the diagnostic performance of Panbio™ and RT-PCR jointly for the detection of SARS-CoV-2 in a clinical setting in Addis Ababa, Ethiopia, using BLCM.

## Materials and methods

### Study design, period, and settings

A health facility-based prospective cross-sectional diagnostic test evaluation study was conducted from July to September 2021, during the third wave of the epidemic, among COVID-

19 presumptive clients in public health facilities of Addis Ababa, Ethiopia. The study site, Addis Ababa is described in Sisay et al., 2022 [30]. As part of the current pandemic response, the Addis Ababa Health bureau selected 20 health centers, i.e., two health centers from the ten sub-cities. The selection is based on the regional health bureau's previous quarter target performance and implementation of the rapid antigen test, Panbio™ COVID-19 Ag Rapid test Device (Abbott Rapid Diagnostics Jena GmbH, Germany) during the pilot phase for the detection of SARS-CoV-2 using nasopharyngeal swabs. From these sites, we have selected two government health centers (Kazanches, Kotebe), and two hospitals (Zewuditu Memorial Hospital, and Ras Desta Damitew Memorial Hospital). These two public hospitals are amongst the largest and the referral health care system promotes and provides preventive, curative and rehabilitative outpatient care including basic laboratory services [32].

## Sampling method and study population

The study population were all presumptive COVID-19 clients among the four public health facilities of Addis Ababa who were willing to take part in the study and were available during the data collection period until the allocated proposed sample to meet as stratified into these four selected sites. We employed a convenience sampling technique. Eligible participants from community surveillance, contacts of confirmed cases, and suspects who fulfill the WHO criteria and Ethiopian guidelines for COVID-19 cases were screened by trained professionals as a quick triage system [15, 33]. Accordingly, a total of 438 clients were enrolled in this study.

We excluded the critically ill cases, confirmed COVID-19 positive clients, and patients younger than 18 years. The reason for the exclusion of confirmed cases was to comply with the criteria for valid diagnostic test studies by including among a consecutive series of patients suspected (but not known) to have the target disorder [34].

## Sample collection and laboratory testing procedures

We collected two nasopharyngeal respiratory specimens from each study participant under strict bio-safety measures using two milliliters of VTM (China, Miraclean Technology Co., Ltd., www.mantacc.com). The Panbio™ tests (Panbio™ Abbott Diagnostic GmbH, Germany) were analyzed immediately according to the manufacturer's instructions. The other collected specimen was packed by a triple packing system for maintaining the safety measures and shipped immediately to Addis Ababa Public health research and emergency management center laboratory (AAPHEML) for RT-PCR testing.

**RT-PCR SARS-CoV-2 testing.** The RNA extraction from all nasopharyngeal samples was performed by a Bioer nuclear extraction automated nucleic acid purification extraction machine (Hangzhou Bioer Technology Co., Ltd. Zhejiang, China) with MgaBio plus virus RNA purification kit II [30]. In all extraction procedures, as part of assuring the quality management system, positive and negative quality controls were incorporated. The SARS-CoV-2 RT-PCR assay was conducted with the BGI Real-Time Fluorescent RT-PCR Kit as described by Sisay et.al, and in the manufacturer instruction (China) [30, 35] using Sansure Biotech MA-6000 (Changsha, China) according to standard operating procedures and the manufacturer's instruction [35]. The assay detects a specific single target gene, which is found in the ORF1ab region of the SARS-CoV-2 genome. The human housekeeping gene β-Actin was the target gene for internal control. The master mixing was done by mixing 20μl master mix reagent and 10μl of the extracted sample RNA to the well pre-filled with PCR-Mix in the following order: no template (negative) control, patient specimen(s), and positive control. For RT-PCR the cut-off was a cycle threshold (Ct) < 38., if the Ct value of the housekeeping gene was not higher than 32 at VIC/HEX and sample [35]. The RT-PCR laboratory results were interpreted as

positive and negative based on the cut-off Ct values of the manufacturer's recommendation. Clients' results were communicated based on the national result reporting channel and only conclusive results of RT-PCR were returned to the participants.

**Ag-RDT SARS-CoV-2 testing.** The collected nasopharyngeal swabs were processed immediately on site using the Panbio™ COVID-19 AG Rapid Test (Abbott Diagnostic GmbH, Germany), which was the only locally available test kit listed by WHO and authorized by the Ethiopian regulatory body for laboratory utilization during the study period [15, 33, 36]. The samples from the swabs were mixed with approximately 300 μl of buffer, and then 5 drops were dispensed into the device. The results were interpreted in the following 15–20 minutes. The test detects the presence of the nucleocapsid (N) proteins of the virus using an immune chromatography assay. For a positive result with the Panbio™, a visible red line must form in the result (T) and a control (C) line. We report as negative when the red line is only present in the control line (c). The presence of a red control line was a prerequisite for a valid test result [36]. The test was performed as per Panbio™ manufacturer recommendation and in vitro diagnostic rapid test for the qualitative detection of SARS-CoV-2 antigen (Ag) [36].

## Data quality assurance

Data compilers and laboratory workers got appropriate orientation on how to assure valid data using the tool and additional written guides have been provided to them on interpreting each of the study variables. The principal investigators have closely supervised the data collection process so as to ensure the completeness and consistency of the data collection. In addition, data were double entered to prevent error during data entry via cross-checking and also finally checked and verified prior to analysis., The raw data, including the description of the variables, is available at (https://osf.io/3pmk6/) with DOI 10.17605/OSF.IO/3PMK6.

## Data analysis

Descriptive data of the research was entered and analyzed using SPSS statistical software version 23 and the freely available software package R [37]. The choice of the selected recorded patient data, including socio-demographic and clinical information, was informed by considering relevant literatures [36, 38]. Binomial 95% confidence intervals have been used, which were obtained following Jeffreys approach in the R package DescTools [39]. With the aim to describe the magnitude of the differences in the investigated variables, we decided to present 95% CIs, assuming that non-overlapping CIs indicate statistical differences with p smaller as 0.05 [40]. Cohen's kappa to assess agreement beyond chance was obtained with the R package psych [41]. A value of 1 implies almost perfect agreement and values less than 1 implies less than perfect agreement, with a range of values between 0 and 1 [42].

## Bayesian latent class model (BLCM)

With the aim to obtain diagnostic tests accuracies in the absence of a perfect reference standard, Bayesian latent class models (BLCM) were fit to the data following the approach from Hui and Walter for two tests and four populations with MCMC (Markov chain Monte Carlo) simulation to construct posteriors in JAGS version 4.3.0 [43] using the runjags package [44]. We assume that our model with two tests (T) and four populations (P) is identifiable, since the Hui Walter paradigm of $P \geq \frac{T}{\left(2^{T-1}-1\right)}$ is fulfilled. We also assumed similar sensitivities and specificities in all four populations. The frequencies of the four combinations of dichotomized Panbio™ and RT-PCR results (++; +-; -+; —) in the four populations (the four health facilities), respectively, were modeled with a multinomial distribution. To allow for potential conditional

dependencies, pairwise covariance between sensitivities and specificities of all RT-PCRs were included in separate models. Model selection, i.e., in or exclusion of conditional dependencies was based on the 95% credibility intervals (including 0 or not) and on Bayesian p-values.

We run models with all patients, as well as models separately for patients with and without symptoms. The model code (S1 Data) was obtained with the function "auto huiwalter" of the runjags package [44]. MCMC simulations were conducted with three chains of 50 000 iterations each, a burn-in of 5000 iterations, and a thinning of 10 iterations. Non- informative priors (beta (1,1)) were used for the sensitivities of both tests, the Panbio™ and the four prevalences corresponding to the four populations, i.e., health facilities. The shape parameters for the specificity of the RT-PCR were obtained with beta buster [45] assuming "to be 95% sure that the specificity is greater than 90% with a mode at 99%" as prior information. Convergence was assessed by visual inspection of the trace plots and the potential scale reduction factor (Gelman Rubin statistic) being below 1.1. A sensitivity analysis was performed by using different combinations of minimally (dbeta(1,1)) or weakly informative priors (dbeta(2,1)).

## Ethical consideration

Ethical approval was obtained from IRB of the department of medical laboratory Sciences, College of Health Sciences, Addis Ababa University (reference-MLS/174/21), IRB office of Addis Ababa Health bureau, AAPHREML (Reference-AAHB/4039/,227) and also from Addis Ababa University, College of natural and computational science IRB (IRB-CNCSDO/604/13/ 2021). Additionally, AAPHREML wrote a support letter to the study health facilities. During data collection process, the data collectors informed each study health facility and study participants about the purpose and anticipated benefits of the research and on their full right to refuse, withdraw or completely reject part or all of their part in the study. Written informed consent on the use of data with full anonymity was obtained from the voluntary participants. This work has been done and performed as per Helsinki declaration.

## Results

### Socio-demographic characteristics of the study participants

A total of 438 presumptive clients were identified and enrolled in this study and the majority of them, 239 (54.6%) were females. The mean age of the participants was 36.38 ±14.3 years (min 18, max 84). Three fourth of the study participants (n = 318) had symptoms of COVID-19 and the most often reported clinical symptoms were cough (n = 191), followed by headache (n = 39). For more than half (n = 258) of the participants, the reason for getting tested was due to observing the classic symptoms. Between the two diagnostic tests, based on non-overlapping 95% CIs, there was no significant difference in any of the assessed demographic and clinical variables, in the proportion of the positive test results. For both tests, there were significantly more positive tests for individuals with COVID-19 symptoms compared to no symptoms and for individuals with a contact to a confirmed case compared to individuals without such a contact. The agreement beyond chance, assessed with Cohen's kappa value, was 0.81 [95% CI: 0.76, 0.87]. The demographic data, including also clinical characteristics related to potential COVID-19 infection, as well as co-morbidities, are presented in Table 1.

### Test results in relation to clinical onset

The majority of samples originate from patients during the first seven days after the onset of clinical symptoms. The highest proportion of positive tests, for both tests, is seen during four

**Table 1. Demographic characteristics, including clinical characteristics related to a potential COVID-19 infection, of study participants and cross-classified results of RT-PCR and Panbio™, Addis Ababa, Ethiopia, 2021 (n = 438).**

| Variable | | n (%) | RT-PCR positive test results n (%) [95% CI] | Panbio™ positive test results n (%) [95% CI] |
|---|---|---|---|---|
| **Gender** | Male | 199 (45.3) | 92 (46.2) [39.4;53.2] | 78 (39.2) [32.6;46.1] |
| | Female | 239 (54.6) | 104 (43.5) [37.2;49.8] | 80 (33.5) [27.7;39.6] |
| **Occupation** | Health worker | 17 (3.9) | 11 (64.7) [41.1;83.7] | 10 (58.8) [35.6;79.3] |
| | Government Employee | 97 (22.1) | 48 (49.59) [39.7;59.3] | 41 (42.3) [32.8;52.2] |
| | Self employed | 63 (14.4) | 29 (46.0) [34.1;58.3] | 28 (44.4) [32.6;56.7] |
| | Private employee | 118 (26.9) | 48 (40.7) [32.1;49.7] | 32 (27.1) [19.7;35.6] |
| | NGO employee | 9 (2.0) | 4 (44.4) [17.3;74.6] | 4 (44.4) [17.3;74.6] |
| | No response /Others | 134 (30.6) | 56 (41.8) [33.7;50.2] | 43 (32.1) [24.6;40.3] |
| **COVID-19 symptoms** | Yes | 318 (72.6) | 164 (51.6) [46.1;57.0] | 139 (43.7) [38.3;49.2] |
| | No | 118 (26.9) | 32 (27.1) [19.7;35.6] | 19 (16.1) [10.3;23.5] |
| | Don't know | 2 (0.4) | 0 (0) [0;6.7] | 0 (0) [0;6.7] |
| **clinical symptoms** | Cough | 191 (43.6) | 107 (56.0) [48.9;62.9] | 91 (47.6) [40.6;54.7] |
| | Fever | 33 (7.5) | 14 (42.2) [26.8;59.3] | 11 (33.3) [19.2;50.3] |
| | Shortness of breath | 13 (3.0) | 6 (46.1) [22.1;71.7] | 6 (46.1) [22.1;71.7] |
| | Sore throat | 21 (4.8) | 7 (33.3) [16.3;54.6] | 6 (28.6) [12.9;49.7] |
| | Headache | 39 (8.9) | 15 (38.5) [24.5;54.1] | 9 (23.1) [12.1;37.9] |
| | Easy fatigue | 6 (1.4) | 3 (50) [16.7;83.3] | 3 (50) [16.7;83.3] |
| | Loss of smell and /or taste | 4 (0.9) | 2 (50) [12.3;87.7] | 2 (50) [12.3;87.7] |
| | Joint &/or muscle pain | 13 (3.0) | 11 (84.6) [59.1;96.6] | 11 (84.6) [59.1;96.6] |
| | 1 to 8(All symptoms) | 2 (0.4) | 2 (100) [33.3;100] | 2 (100) [33.3;100] |
| | No response | 116 (26.5) | 29 (25) [17.8;33.4] | 17 (14.6) [9.1;21.9] |
| **Have comorbidity** | yes | 57 (13.0) | 26 (45.6) [33.2;58.5] | 22 (38.6) [26.8;51.5] |
| | No/no answer | 381 (87.0) | 170 (44.6) [39.7;49.6] | 136 (35.7) [31.0;40.6] |
| **Type of comorbidity** | DM | 26 (5.9) | 13 (50) [31.6;68.4] | 12 (46.1) [28.2;64.9] |
| | Hypertensive | 25 (5.7) | 14 (56) [36.8;73.9] | 10 (40) [22.7;59.4] |
| | HIV/AIDS | 2 (0.4) | 1 (50) [6.1;93.9] | 1 (50) [6.1; 93.9] |
| | Chronic respiratory D/s | 3 (0.7) | 1 (33.3) [3.9;82.3] | 1 (33.3) [3.8;82.3] |
| | Chronic Cardiac D/S | 2 (0.4) | 0 (0) [0;66.7] | 0 (0) [0;66.7] |
| | Malignancy | 2 (0.4) | 1 (50) [6.1;93.9] | 1 (50) [6.1;93.9] |
| | Other and have no comorbidity | 378 (86.3) | 166 (44.1) [39.2;49.2] | 133 (35.2) [30.5;40.1] |
| **Contacts with confirmed case** | Yes | 170 (38.8) | 50 (29.4) [22.9;36.6] | 34 (20) [14.5;26.5] |
| | No | 266 (60.7) | 146 (54.9) [48.9;60.8] | 124 (46.6) [40.7;52.6] |
| | Others | 2 (0.4) | 0 (0) [0;66.7] | 0 (0) [0;66.7] |
| **Assumed place of exposure** | Home | 83 (18.9) | 30 (36.1) [26.4;46.8] | 20 (24.1) [15.9;34.1] |
| | Workplace | 114 (26.0) | 40 (35.1) [26.8;44.1] | 31 (27.2) [19.7;35.8] |
| | Health facility | 1 (0.2) | 0 (0) [0;85.3] | 0 (0) [0;85.3] |
| | Others | 3 (0.7) | 0 (0) [0;53.5] | 0 (0) [0;53.5] |
| | Not recognized | 237 (54.1) | 126 (53.2) [46.8;59.4] | 107 (45.1) [38.9;51.5] |
| **Previously tested positive** | Yes | 60 (13.7) | 23 (38.3) [26.8;50.9] | 14 (23.3) [14.0;35.1] |
| | No | 378 (86.3) | 173 (45.8) [40.8;50.8] | 144 (38.1) [33.3;43.1] |
| **COVID-19 vaccination** | Yes | 49 (11.2) | 17 (34.7) [22.5;48.6] | 11 (22.4) [12.5;35.5] |
| | No | 389 (88.8) | 179 (46.0) [41.1;51.0] | 147 (37.8) [33.1;42.7] |
| **Wear face mask regularly** | Yes | 426 (97.3) | 187 (43.9) [39.2;48.6] | 151 (35.4) [31.0;40.1] |
| | No | 6 (1.4) | 4 (66.7) [28.6;92.3] | 4 (66.7) [28.6;92.3] |
| | No response | 6 (1.4) | 5 (83.3) [44.2;98.1] | 3 (50) [16.7;83.3] |

*(Continued)*

**Table 1.** (Continued)

| Variable | | n (%) | RT-PCR positive test results | Panbio™ positive test results |
|---|---|---|---|---|
| | | | n (%) [95% CI] | n (%) [95% CI] |
| Reason for testing | Suspect | 258 (58.9) | 140 (54.3) [48.2;60.3] | 120 (46.5) [40.5;52.6] |
| | Contact of confirmed case | 177 (40.4) | 56 (31.6) [25.1;38.7] | 38 (21.5) [15.9;27.9] |
| | Community surveillance | 3 (0.7) | 0 (0) [0;53.5] | 0 (0) [0;53.5] |
| Health facility | Zewditu Memorial hospital (HF1) | 46 (10.5) | 23 (50) [35.9;64.1] | 20 (43.5) [29.9;57.8] |
| | Ras Desta Damtew Memorial Hospital (HF2) | 230 (52.5) | 86 (37.4) [31.3;43.8] | 63 (27.4) [21.9;33.4] |
| | Kazenchis Health Center (HF3) | 116 (26.5) | 62 (53.4) [44.4;62.3] | 57 (49.1) [40.1;58.2] |
| | Kotebe Health Center (HF4) | 46 (10.5) | 25 (54.3) [40.0;68.1] | 18 (39.1) [26.0;53.4] |

to seven days after clinical onset. After that, the proportion of positive Panbio™ test results declines earlier compared to RT-PCR. Most of the Ct values were below 30, Table 2.

**Diagnostic performance of Panbio™ and RT-PCR using BLCM.** BLCMs were performed to estimate diagnostic test accuracies of both tests under evaluation, RT-PCR and Panbio™, without assuming the existence of a reference standard. The models considered four populations (the four health centers). Based on the visual inspection of the trace plots and the potential scale reduction factors, being below 1.1 for all parameters of interest, the Markov chain Monte Carlo (MCMC) chains converged. The sensitivity of the Panbio™ was with 99.6 [98.4; 100] % considerably higher than the RT-PCR 89.3 [83.2; 97.6] %. The credibility intervals of the RT-PCR were also wider than those of the Panbio™. The specificity of the RT-PCR was close to being perfect with 99.1 [97.5; 100] % and higher compared to the specificity of the Panbio™ COVID-19 Ag Rapid Test with 93.4 [82.3; 100] %. The posterior estimates and their 95% credibility intervals are presented in Table 3.

Next to models with all patients, two separate models for patients with and without symptoms were run. While the sensitivity of the Panbio™ was similar in patients with and without symptoms, the specificity was considerably lower in patients without symptoms. In contrast, sensitivity and specificity of RT-PCR was only reduced by 1 or 2% in patients without symptoms.

Moreover, a sensitivity analysis was performed using weakly informative prior (dbeta(2,1)) indicating that the prior of RT-PCR did not affect the posterior. Since the posterior credibility intervals of the both covariance terms (conditional dependency between sensitivities or specificities) did include zero and the value of the Bayesian p-value provided no evidence of conditional dependencies, no covariance term was included in the final model, (S2-S6 in S1 File). The posterior estimates of the models with covariance terms are presented in S7 in S1 File.

**Table 2. RT-PCR and Panbio™ test results in relation to days since clinical onset, 2021, Addis Ababa, Ethiopia.**

| Date of clinical onset | RT PCR test result | | | RT PCR positive test result along with Ct. values | | | | Panbio™ Ag RDT result | | |
|---|---|---|---|---|---|---|---|---|---|---|
| | Positive | Negative | Total | Ct ≤25 | Ct>25 to ≤30 | >30 to Ct ≤35 | >35 to Ct <38 | Positive | Negative | Total |
| 0-3days | 63 | 75 | 138 | 50 | 9 | 1 | 3 | 56 | 82 | 138 |
| 4-7days | 100 | 55 | 155 | 66 | 27 | 4 | 3 | 82 | 73 | 155 |
| 8–10 days | 16 | 9 | 25 | 8 | 6 | 2 | 0 | 11 | 14 | 25 |
| 11–15 days | 6 | 6 | 12 | 1 | 3 | 0 | 2 | 3 | 9 | 12 |
| >15 days | 2 | 30 | 32 | 0 | 1 | 0 | 1 | 1 | 31 | 32 |
| No response* | 9 | 67 | 76 | 2 | 3 | 2 | 2 | 5 | 71 | 76 |
| Total | 196 | 242 | 438 | 127 | 49 | 9 | 11 | 158 | 280 | 438 |

*Under "No response" are patients listed which had no clinical symptoms, did not know or were not able to indicate the data of clinical onset.

**Table 3. Performance of test kits using all model of BLCM, Addis Ababa, Ethiopia, 2021.**

| Parameter | Model with all patients Median [95% CrI] | Model: Patients with symptoms Median [95% CrI] | Model:Patients without symptoms Median [95% CrI] |
|---|---|---|---|
| Se_Panbio | 99.6 [98.4;100] | 99.3 [97.4;100] | 99.2 [96.6;100] |
| Se_PCR | 89.3 [83.2;97.6] | 91.4 [83.8;100] | 89.4 [81.3;98.0] |
| Sp_Panbio | 93.4 [82.3;100] | 91.6 [82.7;100] | 83.6 [58.6;100] |
| Sp_PCR | 99.1 [97.5;100] | 99.0 [97.2;100] | 98.0 [93.1;100] |
| Prev Hf1 [1] | 54.0 [39.3;68.4] | 50.0 [33.5;66.2] | 64.4 [36.3;89.1] |
| Prev Hf2 | 70.0 [62.1;77.1] | 58.9 [49.6;68.0] | 90.5 [80.7;97.4] |
| Prev Hf3 | 49.1 [39.4;58.5] | 46.0 [35.5;56.4] | 57.8 [37.7;76.5] |
| Prev Hf4 | 55.2 [38.7;71.2] | 43.3 [25.3;62.3] | 76.8 [51.7;95.6] |

[1]: Prevalence of the health facility (HF). The four health facilities are considered as the four populations in the model.

CrI: Credible interval.

## Discussion

Accurate and reliable diagnostic tests play a crucial role in curbing COVID-19 infection. Accordingly, this study assessed the agreement and the diagnostic performance of the Panbio™ test jointly together with the RT-PCR for the detection of SARS-CoV-2 in a clinical setting using BLCM. The agreement was excellent with 0.81 but not perfect [41, 42].

If RT-PCR is considered as a perfect reference standard (which is highly questionable), the sensitivity of the Panbio™ would be 80.1[74.1; 85.2] % and the specificity 99.6 [98.1, 99.9] %. This is concordant with the findings a study done by Akingba and colleagues in South Africa and Bulilete *et. al.*, in Spain [46, 47].

In contrast, when using a no reference standard approach with BLCM, which is a more realistic approach, the RT-PCR still performed very well with a near perfect specificity of 99.1 [97.5; 100] % and a high, but not perfect sensitivity of 89.3 [83.2; 97.6] % which is comparable finding with Staerk-Østergaard *et. al.*, as the specificity the two candidate tests get a specificity of greater than >99.7%, while their sensitivities are less matched [25].

Unexpectedly, the diagnostic test accuracy of Panbio™ was found to have a very high sensitivity of 99.6 [98.4; 100] % and slightly lower specificity of 93.4 [82.3; 100] %. There are very few studies using BLCM to assess the performance of RDT and to our knowledge there is none that assessed Panbio™, which renders comparisons with published findings difficult. A notable exception is the study from Staerk-Østergaard *et. al.*, [25] which also obtain very high estimates for the specificities of RT-PCR, but with 95[92.8; 98.4] % for RT-PCR sensitivity and 53.8 [49.8; 57.9] % for RDT sensitivity, considerably different values compared to our findings. A number of reasons may explain these differences to our findings. The study from [25] used a huge data set from the National Danish registry including tests results from several rapid antigen tests, which makes a comparison with our study—using a single rapid antigen test and a single RT-PCR assay—difficult. It is also well possible that the Danish patient sample from Staerk-Østergaard *et. al.*, [25], differs from our Ethiopian sample, i.e., the apparent prevalences are considerably higher (more than 40%) in the Ethiopian health centers, compared to the situation in the Danish study with the highest median prevalence 0f 2.56% [25]. In diagnostic test theory, it is well known that diagnostic test accuracies, possibly differ in different populations with different underlying demographic characteristics [48].

The authors of Staerk-Østergaard *et. al.*, [25] describe the situation in Denmark with "a heavy use of antigen testing in primary schools, high schools, and universities". This probably

entails that the large majority of Danish samples in the national registries originate from persons without symptoms. Staerk-Østergaard et al., discuss their relatively low sensitivity compared to the results of another BLCM study [49] with 68.1%, with the latter one not taken in a clinical setting, but in a laboratory one. This sensitivity [49] even increased to 78.8% in samples from symptomatic patients. In our study, nearly three-quarters of the patients reported symptoms compatible with COVID-19, which might explain the unexpectedly high sensitivity.

In a number of diagnostic test evaluation studies, it became evident, that the number of false negative tests in RDT increases with time after the onset of clinical symptoms and low Ct value, especially after more than 1–2 weeks [17–19], while the sensitivity of RT-PCR still remains high. In our sample, there were just 15.7% of the samples originated from patients taken after 8 days of clinical onset. Additionally, 17.3% of the patients could not indicate the data of clinical onset possibly some of them had no clinical symptoms. These demographic characteristics of our patient sample—a high proportion tested within the first ten days after clinical onset—might be another explanation for the high sensitivity of the Panbio™ test.

Due to the high proportion of patients with symptoms tested within the first ten days after clinical onset, as a limitation, we cannot generalize our findings to patients being tested later in the course of the disease. Additionally, our study was not designed to assess the effect of the presence/absence of symptoms on diagnostic test accuracies, with considerably more patients with symptoms. Therefore, this findings needs to be interpreted with caution.

Both tests showed comparable diagnostic accuracies in patients with and without symptoms, with the exception of the specificity of Panbio™, which was slightly lower in asymptomatic patients. With regard to the high sensitivity, health professionals could use this rapid antigen test kit for screening clients in particular in patients with symptoms within a few days after clinical onset.

The RDT testing was performed by well trained personnel under strict biosafety control, thus our results of the RDT pertain to testing under these conditions, which might not be attainable in the whole country if Panbio™ is widely applied in different settings.

In the present study, we excluded the critically ill clients as they lack decisional capacity due to their clinical status and also we are unable to get sufficient nasopharyngeal swabs from them, thus the results of our study are strictly speaking only generalizable to the study population. Since there were just minor differences in the sensitivities between patients with and without symptoms, we assume that it is very likely that also critically ill COVID-19 patients would be correctly classified by the Panbio™. Furthermore, possibly the low number of fully vaccinated patients enrolled in our cases might differ compared to other settings [50–52].

Our study findings are in line with the WHO stated criteria for the emergency use of COVID-19 diagnostic tests considered as a replacement for laboratory-based RT-PCR in the clinical setting (sensitivity ≥ 80% and specificity ≥ 97%), at least for the sensitivity. With regard to the lower specificity of Panbio™, a positive test result, especially in asymptomatic patients has important consequences, and a subsequent, confirmatory RT-PCR is warranted [11, 53, 54].

Due to a lack of resources, we were not able to monitor the viral load quantitatively, nor to determine the genetic variants of SARS-CoV-2 present in our samples. This information might have provided more insights and further explanations of our findings.

With BLCM, our study uses a statistical approach, which is still novel to the medical field. We suggest that for future pandemic preparedness, similar to regularly conducted proficiency testing (ring testing) to control analytical sensitivity and specificity, an approach to monitor field performance of newly developed diagnostic tests is developed. This would entail having access to appropriate patient samples and associated data, as well as an exchange on BLCM methods.

## Conclusion

To our knowledge, our study is the first prospectively designed study to assess diagnostic test accuracies of Panbio™ (or any RDT) with BLCM in a clinical setting. Additionally, this study took place in a low-income country, where information on diagnostic test accuracies is scarce and equity in access to health services is not guaranteed. Here, RDT due to its lower costs and ease of application, is a valid alternative to RT-PCR.

From a clinical perspective, in case of doubt, i.e. in asymptomatic individuals and if a false positive test result would have important unwanted consequences (quarantine), a confirmatory RT-PCR test with a near perfect specificity is warranted.

Based on our results, with its high sensitivity and an acceptable specificity, the Panbio™, provides a viable alternative to RT-PCR for detecting COVID-19 patients.

## Supporting information

**S1 Data. BLCM.** The model code.
(R)

**S1 File.**
(DOCX)

## Acknowledgments

We are very grateful and acknowledge the Addis Ababa University and Addis Ababa Health Bureau, the Zewditu Memorial Hospital, the Ras Desta Damtew Memorial Hospital, the Kotebe Health Center, and the Kazanches Health Center for granting their institutional ethical approval and their support and assistance in accessing diverse resources used in the study.

We also acknowledge all the study participants.

## Author Contributions

**Conceptualization:** Abay Sisay, Adey Feleke Desta.

**Data curation:** Abay Sisay, Abebaw Tiruneh, Yasin Desalegn, Abraham Tesfaye, Adey Feleke Desta.

**Formal analysis:** Abay Sisay, Sonja Hartnack.

**Funding acquisition:** Abay Sisay.

**Investigation:** Abay Sisay, Sonja Hartnack, Abebaw Tiruneh, Yasin Desalegn, Abraham Tesfaye, Adey Feleke Desta.

**Methodology:** Abay Sisay, Sonja Hartnack, Abebaw Tiruneh, Yasin Desalegn, Abraham Tesfaye, Adey Feleke Desta.

**Project administration:** Abay Sisay.

**Resources:** Abay Sisay.

**Software:** Abay Sisay, Sonja Hartnack, Adey Feleke Desta.

**Supervision:** Abay Sisay, Sonja Hartnack, Abebaw Tiruneh, Abraham Tesfaye, Adey Feleke Desta.

**Validation:** Abay Sisay, Sonja Hartnack, Abebaw Tiruneh, Abraham Tesfaye, Adey Feleke Desta.

**Visualization:** Abay Sisay, Sonja Hartnack, Adey Feleke Desta.

**Writing – original draft:** Abay Sisay, Sonja Hartnack, Abebaw Tiruneh, Yasin Desalegn, Abraham Tesfaye, Adey Feleke Desta.

**Writing – review & editing:** Abay Sisay, Sonja Hartnack, Abebaw Tiruneh, Yasin Desalegn, Abraham Tesfaye, Adey Feleke Desta.

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
