## [Decision Letter · Decision Letter 0]

17 May 2022

PONE-D-22-11748Evaluating diagnostic accuracies of Panbio™ COVID-19 rapid antigen test and RT-PCR for the detection of SARS-CoV-2 in Addis Ababa, Ethiopia using Bayesian Latent-Class Models (BLCM)PLOS ONE

Dear Mr. Sisay,

Thank you for submitting your manuscript to PLOS ONE. After careful consideration, we feel that it has merit but does not fully meet PLOS ONE’s publication criteria as it currently stands. Therefore, we invite you to submit a revised version of the manuscript that addresses the points raised during the review process.

We look forward to receiving your revised manuscript.

Kind regards,

A. K. M. Anisur Rahman, Ph.D.

Academic Editor

PLOS ONE

Journal Requirements:

Reviewers' comments:

Reviewer's Responses to Questions

**Comments to the Author**

1. Is the manuscript technically sound, and do the data support the conclusions?

Reviewer #1: Partly

Reviewer #2: Partly

2. Has the statistical analysis been performed appropriately and rigorously? 

Reviewer #1: Yes

Reviewer #2: I Don't Know

3. Have the authors made all data underlying the findings in their manuscript fully available?

Reviewer #1: Yes

Reviewer #2: No

4. Is the manuscript presented in an intelligible fashion and written in standard English?

Reviewer #1: No

Reviewer #2: No

5. Review Comments to the Author

Reviewer #1: Abstract

Results: add percentage symbol (%)

Keywords: revise “no gold standard”

Introduction

Revise lines 68 – 76.

Line 99: Introduce BLMC acronyms

Need an explanation of BLMC for diagnose. Advantage and disadvantages.

Material and methods

Line 112: add manufacturer, city and country of Panbio RDTs

Line 128: the correct header is Study population

Line 137: remove “methods”

Line 142: add information of manufacturer, city and country for MTVs

Lines 146 -149: revise the phrases. It is not clear.

Lines 104 – 126: Combine the Study design and settings with sampling method

Put together the sample collection into the laboratory testing procedures.

Lines 152 – 156: add information of manufacturer, city and country for the RNA extractor, PCR analyzer and detection kits.

Lines 162 – 164: Revise the phrase

Lines 174 – 178: Revise the phrases. Additionally, is not clear if the Ag-RDTs were nasal or nasopharyngeal swabs based.

Line 196 – remove the [

Line 22 – remove )

In the method chapter is not described which result returned to the participants. Please add it.

Results

The explanation of results is missed. Please add it before each table.

Add % for percentual numbers.

Lines 242 – 243: start with mean and add in ( ) the min and max of age.

Table 1: Add horizontal lines for each variable.

Organize the numbers: put n (%) up and 95% CI above n (%).

Lines 266 – 268: revise the phrase.

Harmonized if is Ct or ct value.

Lines 274 – 278: Revise the phrase.

Discussion

The discussion chapter must be revised. The meaning of the phrases is not clear.

Do not repeat the results but discuss it.

Reviewer #2: This primary study aims to evaluate the accuracy of both a rapid antigen test (Panbio) and RT-PCR for the detection of SARS-CoV-2 using a Bayesian latent class model. The model used is in contrast to the standard DTA paradigm in which a test such as Panbio is compared with a reference standard result RT-PCR. Although not a new approach, BLCMs are still not in frequent use in the test evaluation literature and unfortunately I do not think the authors have done enough here to either justify the approach taken or to explain the model used. I have highlighted particular areas of concern and made some suggestions below that the authors may wish to consider.

Introduction:

This section should introduce the BLCM and why it has been selected for use in this study. As mentioned above it is not yet a standard approach to analysis of test accuracy and therefore the reader needs to better understand the context. The basis on which PCR is claimed to be an ‘imperfect’ reference standard is not mentioned at all.

Line 82 – Rapid antigen tests are claimed to have comparable diagnostic performance to RT-PCR but as a general rule this is just not true. RATs do seem to have equivalently high specificity compared to PCR but sensitivity for detecting SARS-CoV-2 is on average considerably lower, unless the authors mean this sentence to refer to test performance in individuals with higher viral load.

Line 87 – It is rather an overstatement to claim that technical personnel are ‘prone’ to contracting the virus

Line 93 – It is just not true to state that there is no documented evidence on the performance of Panbio. It is one of the most studies antigen tests that are on the market.

Line 98 – use of PCR as a reference standard ‘inevitably leads to bias’ – why and in what direction? The target condition for this study appears to be detection of SARS-CoV-2 infection (rather than detection of infectious cases, for example), in which case PCR is usually considered to be an adequate, if not perfect, reference standard.

Methods

The participant recruitment is generally well described however the relevance of the ‘four populations’ from the four study sites could be signposted here to avoid later confusion. A PRISMA style participant flow diagram would be useful to describe the participant flow through the study and any exclusions that were made.

Line 139 – Implies both nasopharyngeal samples were collected in VTM before one was used with the rapid antigen test. This would be in contradiction to the manufacturer instructions for this test which recommends direct testing, and from subsequent text, I suspect may not be what actually happened. Please clarify

Line 197 – I can’t understand why Cohen’s kappa is used to assessment the agreement between RAT and PCR. It requires a cross-tabulation of the two test results so why not present that with the 5 results that disagree +/- and -/+ so that the reader can instantly see where the disagreement lies?

Line 200- BLCM model – Considerably more information is needed here to explain what was done and the assumptions made in this model. On what basis was PCR considered imperfect and how was this dealt with by the model. Why were four populations modelled? I am not a statistician but I believe there is insufficient detail here even for a statistician to fully follow the approach taken.

Results

Table 1 – why have 95% CIs been reported for the percentages reported in the Table? It makes for a very messy read and the relevance is not clear to me.

Table 2 – what does the line ‘No response’ refer to – is it lack of symptoms?

Table 3a – the results presented are completely the opposite to what one would intuitively expect – Panbio is reportedly 10 percentage points more sensitive than PCR and considerably less specific – however no explanation of this finding, why it came about and what it means is given either in Results or Discussion.

Table 3a - What does Prev HF1, Prev HF2 etc refer to and how does this influence the model?

Table 3b – the caption states ‘Performance of test kits on conditional dependency using BLCM’ – does this mean that the baseline model did not assume conditional dependency between the two tests? This would be an odd assumption given the target condition and the way in which the two tests work (detection viral protein and viral RNA are bound to be correlated particularly in the early phase of infection when most of these samples were taken).

Discussion

This section focuses on comparing study results to selected studies analysed under a more standard DTA framework rather explaining and justifying the rather unusual results obtained here. I ma not opposed to the approach taken but the authors need to do a lot more to put this study and their results in context.

There are some grammatical concerns that need to be addressed throughout the paper, e.g.

Line 62 – ‘laboratory diagnostic capacities are at their questionable’

Line 64 – ‘realizing the 2030 SDG with lots of pitfalls in the diagnostic capacity’

Line 78 – ‘its affordability and infrastructure demand for most laboratories in the low-income countries are atypical’

Line 106 – ‘during on the third wave’

Line 266 – Sentence beginning ‘As the ct values of the test …’ does not make any sense.

6. PLOS authors have the option to publish the peer review history of their article (what does this mean?). If published, this will include your full peer review and any attached files.

Reviewer #1: No

Reviewer #2: **Yes: **Jacqueline Dinnes

---

## [Author Response · Author response to Decision Letter 0]

4 Jul 2022

Dear Sir/ Madam

We are delighted to have an opportunity to make our manuscript revised and we have greatly appreciated the reviewers’ high-level comments, and suggestions were very helpful for the overall improvement of the manuscript. 

Accordingly, we did a substantial change and improvement on the manuscript. 

As instructed, we have attempted to succinctly explain changes made in reaction to all comments and reply to each comment in a point-by-point fashion as submitted it ”Response to reviewers” through the system.

Please kindly have it.

---

## [Decision Letter · Decision Letter 1]

2 Aug 2022

PONE-D-22-11748R1Evaluating diagnostic accuracies of Panbio™ COVID-19 rapid antigen test and RT-PCR for the detection of SARS-CoV-2 in Addis Ababa, Ethiopia using Bayesian Latent-Class Models (BLCM)PLOS ONE

Dear Dr. Sisay,

Thank you for submitting your manuscript to PLOS ONE. Both of the reviewers have raised some important issues after reviewing the revised version of the manuscript. Therefore, we invite you to submit a revised version of the manuscript that addresses the points raised during the review process.

We look forward to receiving your revised manuscript.

Kind regards,

A. K. M. Anisur Rahman, Ph.D.

Academic Editor

PLOS ONE

Reviewers' comments:

Reviewer's Responses to Questions

**Comments to the Author**

1. If the authors have adequately addressed your comments raised in a previous round of review and you feel that this manuscript is now acceptable for publication, you may indicate that here to bypass the “Comments to the Author” section, enter your conflict of interest statement in the “Confidential to Editor” section, and submit your "Accept" recommendation.

Reviewer #1: (No Response)

Reviewer #2: (No Response)

2. Is the manuscript technically sound, and do the data support the conclusions?

Reviewer #1: Yes

Reviewer #2: Partly

3. Has the statistical analysis been performed appropriately and rigorously? 

Reviewer #1: Yes

Reviewer #2: I Don't Know

4. Have the authors made all data underlying the findings in their manuscript fully available?

Reviewer #1: No

Reviewer #2: Yes

5. Is the manuscript presented in an intelligible fashion and written in standard English?

Reviewer #1: No

Reviewer #2: Yes

6. Review Comments to the Author

Reviewer #1: The manuscript needs to count a history that is easy to understand the begin, middle and end. Please revise the manuscript accordingly, because that it is an important study.

Methods

There are phrases repetition regarding the sample collection in the sample collection and PCR testing procedures.

Line 247 - check if the nasopharyngeal swabs were collected twice or the authors collected 2 NP swabs

Add the manufacturers for all kits and equipment.

Lines 257 - 258, must be moved to data analysis or statistical analysis chapter

Lines 252 and 262 is not clear if the authors collected sample to 2 or 3 ml of VTM.

Lines 292 - 295 should be stated before the results interpretation.

Lines 295-296 must said that only conclusive results of RT-PCR returned to the participants.

Results

Table 1 described demographic but also clinical characteristics of study participants. Please, add that.

The results were not fully described, specially the tables 2 and 3.

Discussion

Need to add the study limitations and discuss the novelty of the study or the confirmation of the existed studies.

Part of conclusions and recommendation has to be in the discussion section.

Reviewer #2: 1. The introduction has been improved with a new paragraph describing BLCM and why it has been selected for use in this study, with some additional information also provided in the Methods however I believe more could be done to help the reader understand and put the results in context. The overall sensitivity of Panbio was found to be 99.6% which is incredibly high considering the sample included patients who were 1, 2 or more weeks after onset of symptoms, and there is abundant evidence of false negative antigen test results in people with Covid-19 especially with increasing time from onset. In their response the authors state that they address the apparently counter-intuitive findings for antigen tests vis-à-vis PCR but I’m afraid I can’t find it anywhere. It is not that I disagree with the use of the BLCM approach but would like to see more done to explain the model assumptions, how results should be interpreted and what might have led to the observed results (other examples in the literature are much easier to follow, e.g. https://onlinelibrary.wiley.com/doi/full/10.1002/jmv.27943).

2. I do not disagree that PCR is imperfect nor that accuracy varies between different PCR assays, however this does not make it true to state that RATs have comparable diagnostic performance to PCR (line 77). Notably this statement is supported by three references, by one comment piece, one non-systematic review of laboratory tests and a primary study of PCR. Because PCR detects viral RNA, then by definition it will detect more cases of SARS-CoV-2 and for a longer time period compared to an antigen test that relies on the presence of sufficient viral antigen in the collected sample. This does not mean that PCR has no false negative results, nor that it is always the best test to use, however if an antigen test is picking up cases of SARS-CoV-2 that are missed by PCR (i.e. Ag is more sensitive than PCR) then under a standard DTA paradigm one would surely expect to see more false positive rapid antigen test results?

3. The authors suggest that lines 136-140 are sufficient to signpost the use of ‘four populations’ however I disagree and it would not take much to add a statement to the effect that the four sites used are henceforward referred to as ‘four populations’ contributing to the model.

4. The author suggest that 95% CIs reported in Table 2 are helpful to indicate the uncertainty of the sample and to identify statistically significant differences between groups. I still do not think a 95% CI for each covariate overall is very helpful (first column) and for the comparison of results between PCR positive and Panbio positive participants, surely it is the 95% CI of the difference in mean between the two groups that is relevant rather than each 95% CI per group?

5. Please add a footnote to Table 2 to define ‘No response’ (i.e. clients did not know and were not able to recognize the date of clinical onset.) and one to Table 3a to define Prev HF1, Prev HF2 etc (i.e. these indicate the four different health facilities)

6. Author state that they have revised the Discussion to explain and justify the counter-intuitive results obtained but I cannot see any such revisions.

7. PLOS authors have the option to publish the peer review history of their article (what does this mean?). If published, this will include your full peer review and any attached files.

Reviewer #1: No

Reviewer #2: No

---

## [Author Response · Author response to Decision Letter 1]

7 Sep 2022

Dear esteemed reviewers

Once again, it’s a good opportunity to make our manuscript revised and we have greatly appreciated the reviewers’ high-level comments, and suggestions were very helpful for the overall improvement of the manuscript. 

Accordingly, we did substantial changes to the manuscript following your comments in way of a point-by-point response

Regards!

Abay

---

## [Decision Letter · Decision Letter 2]

21 Sep 2022

PONE-D-22-11748R2Evaluating diagnostic accuracies of Panbio™ test and RT-PCR for the detection of SARS-CoV-2 in Addis Ababa, Ethiopia using Bayesian Latent-Class Models (BLCM)PLOS ONE

Dear Dr. Sisay,

Thank you for submitting your manuscript to PLOS ONE. After careful consideration, we feel that it has merit but does not fully meet PLOS ONE’s publication criteria as it currently stands. Therefore, we invite you to submit a revised version of the manuscript that addresses the points raised during the review process.

ACADEMIC EDITOR:

Line 231: Please delete "The raw data is available" as this was repeatedLines 261-262: Please replace "Minimally informative priors" with "non-informative priors" as beta[1,1] indicates that.Please remove 95% CI from the first column of Table 1 to make more readable without losing important information.Please elaborate CrI in table 3a footnotePlease remove Table 3b from main text and present as a supplementary file as conditional dependence was not considered in the final modelLine 431: Please delete "and recommendations"Line 432: Please replace "our study" with "this"Lines 442-450: Please delete these sentences or put in an appropirate place under discussion section

Kind regards,

A. K. M. Anisur Rahman, Ph.D.

Academic Editor

PLOS ONE

Journal Requirements:

Additional Editor Comments (if provided):

Reviewers' comments:

Reviewer's Responses to Questions

**Comments to the Author**

1. If the authors have adequately addressed your comments raised in a previous round of review and you feel that this manuscript is now acceptable for publication, you may indicate that here to bypass the “Comments to the Author” section, enter your conflict of interest statement in the “Confidential to Editor” section, and submit your "Accept" recommendation.

Reviewer #2: All comments have been addressed

2. Is the manuscript technically sound, and do the data support the conclusions?

Reviewer #2: Yes

3. Has the statistical analysis been performed appropriately and rigorously? 

Reviewer #2: I Don't Know

4. Have the authors made all data underlying the findings in their manuscript fully available?

Reviewer #2: Yes

5. Is the manuscript presented in an intelligible fashion and written in standard English?

Reviewer #2: Yes

6. Review Comments to the Author

Reviewer #2: The manuscript has been substantially improved since my last review, and my concerns have largely addressed. I particularly appreciate the comparison with results of other BLCM models in the Discussion. A minor request is to replace use of the term 'gold standard' with the recommended 'reference standard'.

7. PLOS authors have the option to publish the peer review history of their article (what does this mean?). If published, this will include your full peer review and any attached files.

Reviewer #2: No

---

## [Author Response · Author response to Decision Letter 2]

24 Sep 2022

Dear Editor: 

We are very much pleased to write this response that our research manuscript entitled “Evaluating diagnostic accuracies of Panbio™ test and RT-PCR for the detection of SARS-CoV-2 in Addis Ababa, Ethiopia using Bayesian Latent-Class Models (BLCM), with a manuscript reference number PONE-D-22-11748R2” has been possibly considered for publication in PLOS ONE. 

We are still greatly appreciated your high-level comments, and suggestions on the manuscript. It’s a good opportunity to make our manuscript more revised.

Accordingly, we did such changes to the manuscript following the Academic editor and reviewer’s comments in way of a point-by-point response

---

## [Editor Report · Decision Letter 3]

5 Oct 2022

Evaluating diagnostic accuracies of Panbio™ test and RT-PCR for the detection of SARS-CoV-2 in Addis Ababa, Ethiopia using Bayesian Latent-Class Models (BLCM)

PONE-D-22-11748R3

Dear Dr. Sisay,

We’re pleased to inform you that your manuscript has been judged scientifically suitable for publication and will be formally accepted for publication once it meets all outstanding technical requirements.

Kind regards,

A. K. M. Anisur Rahman, Ph.D.

Academic Editor

PLOS ONE